# Diversified Dynamic Routing for Vision Tasks

**Abstract.** Deep learning models for vision tasks are trained on large datasets under the assumption that there exists a universal representation that can be used to make predictions for all samples. Whereas high complexity models are proven to be capable of learning such representations, a mixture of experts trained on specific subsets of the data can infer the labels more efficiently. However using mixture of experts poses two new problems, namely (**i**) assigning the correct expert at inference time when a new unseen sample is presented. (**ii**) Finding the optimal partitioning of the training data, such that the experts rely the least on common features. In Dynamic Routing (DR) [21] a novel architecture is proposed where each layer is composed of a set of experts, however without addressing the two challenges we demonstrate that the model reverts to using the same subset of experts. In our method, Diversified Dynamic Routing (DivDR) the model is explicitly trained to solve the challenge of finding relevant partitioning of the data and assigning the correct experts in an unsupervised approach. We conduct several experiments on semantic segmentation on Cityscapes and object detection and instance segmentation on MS-COCO showing improved performance over several baselines.

## 1 Introduction

In recent years, deep learning models have made huge strides solving complex tasks in computer vision, e.g. segmentation [27,4] and detection [10,34], and reinforcement learning, e.g. playing atari games [30]. Despite this progress, the computational complexity of such models still poses a challenge for practical deployment that requires accurate real-time performance. This has incited a rich body of work tackling the accuracy complexity trade-off from various angles. For instance, a class of methods tackle this trade-off by developing more efficient architectures [38,48], while others initially train larger models and then later distill them into smaller more efficient models [15,46,12]. Moreover, several works rely on sparse regularization approaches [41,9,36] during training or by performing a post-training pruning of model weights that contribute marginally to the final prediction. While listing all categories of methods tackling this trade-off is beyond the scope of this paper, to the best of our knowledge, they all share the assumption that predicting the correct label requires a universal set of features that works best for all samples. We argue that such an assumption is often broken even in well curated datasets. For example, in the task of segmentation, object sizes can widely vary across the dataset requiring different computational

effort to process. That is to say, large objects can be easily processed under lower resolutions while smaller objects require processing in high resolution to retain accuracy. This opens doors for class of methods that rely on *local experts*; efficient models trained directly on each subset separately leveraging the use of this local bias. However, prior art often ignore local biases in the training and validation datasets when tackling the accuracy-efficiency trade-off for two key reasons illustrated in Figure 1. (**i**) Even under the assumption that such local biases in the training data are known, during inference time, new unseen samples need to be assigned to the correct local subset so as to use the corresponding *local expert* for prediction (Figure 1 left). (**ii**) Such local biases in datasets are not known **apriori** and may require a prohibitively expensive inspection of the underlying dataset (Figure 1 right).

In this paper, we take an orthogonal direction to prior art on the accuracy-efficiency trade-off by addressing the two challenges in an unsupervised manner. In particular, we show that training *local experts* on learnt subsets sharing local biases can jointly outperform *global experts*, i.e. models that were trained over the entire dataset. We summarize our contributions in two folds.

1. We propose Diversified Dynamic Routing (DivDR); an unsupervised learning approach that trains several local experts on learnt subsets of the training dataset. At inference time, DivDR assigns the correct local expert for prediction to newly unseen samples.
2. We extensively evcaluate DivDR and compare against several existing methods on semantic segmentation, object detection and instance segmentation on various datasets, i.e. Cityscapes [8] and MS-COCO [24]. We find that DivDR compared to existing methods better trades-off accuracy and efficiency. We complement our experiments with various ablations demonstrating robustness of DivDR to choices of hyperparameters.

## 2    Related Work

In prior literature model architectures were predominantly hand-designed, meaning that hyper-parameters such as the number and width of layers, size and stride of convolution kernels were predefined. In contrast, Neural Architecture Search [54,26] revealed that searching over said hyper-parameter space is feasible provided enough data and compute power resulting in substantial improvement in model accuracy. Recently, a line of research [20,25,3,38,40] also proposed to constrain the search space to cost-efficient models that jointly optimize the accuracy and the computational complexity of the models. Concurrently, cost-efficient inference has been also in the focus of works on dynamic network architectures [31,47,42,44], where the idea is to allow the model to choose different architectures based on the input through gating computational blocks during inference.

For example, Li et al. [21] proposed an end-to-end dynamic routing framework that generates routes within the architecture that vary per input sample.

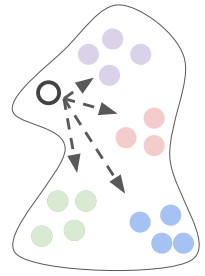 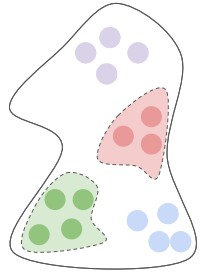

Expert assignment
**during inference**

Finding meaningful subsets
**during training**

Fig. 1: The figure depicts the two main challenges in learning local experts on subsets on subsets of the dataset with local biases. First, even when the subsets in the training dataset is presented where there is a local expert per subset, the challenge remains in assigning the local expert for new unseen samples (left Figure). The second challenge is that the local biases in the training data are not available during training time (right Figure).

The search space of [21], inspired by Auto-DeepLab [25], allows exploring spatial up and down-sampling between subsequent layers which distinguishes the work from prior dynamic routing methods. One common failure mode of dynamic models is mentioned in [31], where early on in the training only a specific set of modules are selected and trained, leading to a static model with reduced capacity. This issue is addressed by Mullapudi et al. [31] through clustering the training data in advance based on latent representations of a pretrained image classifier model, whereas [40] uses the Gumbel-Softmax reparameterization [17] to improve diversity of the dynamic routes. In this work, to mitigate this problem, we adopt the metric learning Magnet Loss [35] which acts as an improvement over metric learning methods that act on the instance level, e.g. Triplet Loss [43,19], and Contrastive Learning methods [7,13]. This is since it considers the complete distribution of the underlying data resulting in a more stable clustering. To adapt Magnet Loss to resolving the Dynamic Routing drawbacks, we use it as an unsupervised approach to increase the distance between the forward paths learned by the Dynamic Routing model this is as opposed to clustering the learned representations, i.e. learning clustered dynamic routes as opposed to clustered representations.

We review the recent advances on semantic segmentation and object detection which are utilized to validate our method in this work. For semantic segmentation, numerous works have been proposed to capture the larger receptive field [49,4,5,6] or establish long-range pixel relation [50,16,37] based on FCN [27]. As mentioned above, with the development of neural network, NAS-based approaches [3,25,32] and dynamic networks [21] are utilized to adjust network architecture according to the data while being jointly optimized to reduce the cost of inference. As for object detection, modern detectors can be roughly divided

into one-stage or two-stage detectors. One-stage detectors usually make predictions based on the prior guesses, like anchors [33,23] and object centers [39,52]. Meanwhile, two-stage detectors predict boxes based on predefined proposals in a coarse-to-fine manner [11,10,34]. There are also several advances in Transformer-based approaches for image recognition tasks such as segmentation [51,45] and object detection [1,53], and while our method can be generalized to those architectures as well, it is beyond the scope of this paper.

## 3    DivDR: Diversified Dynamic Routing

We first start by introducing Dynamic Routing. Second, we formulate our objective of the iterative clustering of the dataset and the learning of experts per dataset cluster. At last, we propose a contrastive learning approach based on *magnet loss* [35] over the gate activation of the dynamic routing model to encourage the learning of different architectures over different dataset clusters.

### 3.1    Dynamic Routing Preliminaries

The Dynamic Routing (DR) [21] model for semantic segmentation consists of $L$ sequential feed-forward layers in which dynamic *nodes* process and propagate the information. Each dynamic node has two parts: (**i**) the *cell* that performs a non-linear transformation to the input of the node; and (**ii**) the *gate* that decides which node receives the output of the cell operation in the subsequent layer. In particular, the gates in DR determine what resolution/scale of the activation to be used. That is to say, each gate determines whether the activation output of the cell is to be propagated at the same resolution, up-scaled, or down-scaled by a factor of 2 in the following layer. Observe that the gate activation determines the *architecture* for a given input since this determines a unique set of connections defining the architecture. The output of the final layer of the nodes are up-sampled and fused by $1 \times 1$ convolutions to match the original resolution of the input image. For an input-label pair $(x, y)$ in a dataset $\mathcal{D}$ of $N$ pairs, let the DR network parameterized by $\theta$ be given as $f_\theta : \mathcal{X} \to \mathcal{Y}$ where $x \in \mathcal{X}$ and $y \in \mathcal{Y}$. Moreover, let $\mathcal{A}_{\tilde{\theta}} : \mathcal{X} \to [0, 1]^n$, where $\theta \supseteq \tilde{\theta}$, denote the gate activation map for a given input, i.e. the gates determining the architecture discussed earlier, then the training objective for DR networks under computational budget constraints have the following form:

$$\mathcal{L}_{DR} = \frac{1}{N} \sum_{i=1}^{N} \mathcal{L}_{seg}\big(f_\theta(x_i), y_i\big) + \lambda \mathcal{L}_{cost}(\mathcal{A}_{\tilde{\theta}}(x_i)). \qquad (1)$$

We will drop the subscript $\tilde{\theta}$ throughout to reduce text clutter. Note that $\mathcal{L}_{seg}$ and $\mathcal{L}_{cost}$ denote the segmentation and computational budget constraint respectively. Observe that when most of the gate activations are sparse, this incurs a more efficient network that may be at the expense of of accuracy and hence the trade-off through the penalty $\lambda$.

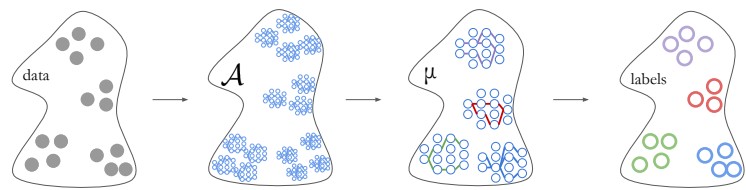

Fig. 2: **Gate Activation cluster assignment.** To update the local experts, DivDR performs K-means clustering on the gate activations over the $\mathcal{A}(x_i)\ \forall i$ in the training examples with fixed model parameters $\theta$.

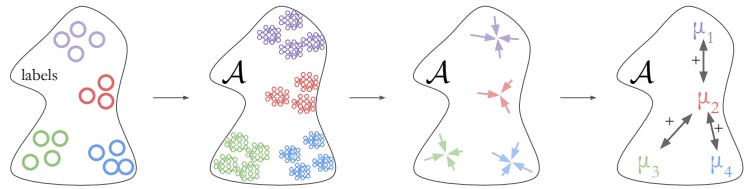

Fig. 3: **Gate Activation Diversification.** We use the labels from the cluster assignment to reduce the *intra-cluster* variance and increase the *inter-cluster* variance by updating model parameters $\theta$.

### 3.2  Metric Learning in $\mathcal{A}$-space

As shown earlier, learning local experts can benefit performance both in terms of accuracy and computational cost. We propose an unsupervised approach to learning jointly the subset of the dataset and the soft assignment of the corresponding architectures. We use the DR framework for our approach.

We first assume that there are $K$ clusters in the dataset for which we seek to learn an expert on each. Moreover, let $\{\mu_{\mathcal{A}_i}\}_{i=1}^K$, denote the cluster centers representing $K$ different gate activations. Note that as per the previous discussion, each gate activation $\mu_{\mathcal{A}_i} \in [0,1]^n$ corresponds to a unique architecture. The set of cluster centers representing gate activations $\{\mu_{\mathcal{A}_i}\}_{i=1}^K$ can be viewed as a set of prototypical architectures for $K$ different subsets in the datasets. Next, let $\mu(x)$ denote the nearest gate activation center to the gate activation $\mathcal{A}(x)$, i.e. $\mu(x) = \arg\min_i \|\mathcal{A}(x) - \mu_{\mathcal{A}_i}\|$. Now, we seek to solve for both the gate activation centers $\{\mu_{\mathcal{A}_i}\}_{i=1}^K$ and the parameters $\theta$ such that the gate activation centers are pushed away from one another. To that end, we propose the alternating between clustering and the minimization of a *magnet loss*[35] variant. In particular, for a given fixed set of activating gates centers $\{\mu_{\mathcal{A}_i}\}_{i=1}^K$, we consider the following loss function:

$$\mathcal{L}_{\text{clustering}}(\mathcal{A}(x_i)) = \left\{ \alpha + \frac{1}{2\sigma^2} \|\mathcal{A}(x_i) - \mu(x_i)\| \right.$$

$$\left. + \log \left( \sum_{k:\mu_{\mathcal{A}_k} \neq \mu(x_i)} e^{-\frac{1}{2\sigma^2} \|\mathcal{A}(x_i) - \mu_{\mathcal{A}_k}\|} \right) \right\}_+ . \quad (2)$$

Note that $\{x\}_+ = \max(x, 0)$, $\sigma^2 = \frac{1}{N-1} \sum_i^N \|\mathcal{A}(x_i) - \mu(x_i)\|^2$, and that $\alpha \geq 0$. Observe that unlike in *magnet loss*, we seek to cluster the set of architectures by separating the gate activations. Note that the penultimate term pulls the architecture, closer to the most similar prototypical architecture while the last term pushes it away from all other architectures. Thereof, this loss incites the learning of $K$ different architectures where each input $x_i$ will be assigned to be predicted with one of the $K$ learnt architectures. To that end, our overall *Diversified* DR loss is given as follows:

$$\mathcal{L}_{\text{DivDR}} = \frac{1}{N} \sum_{i=1}^{N} \mathcal{L}_{segm}(f_\theta(x_i), y_i) + \lambda_1 \mathcal{L}_{cost}(\mathcal{A}(x_i)) + \lambda_2 \mathcal{L}_{clustering}(\mathcal{A}(x_i)). \quad (3)$$

We then alternate between minimizing $\mathcal{L}_{\text{DivDR}}$ over the parameters $\theta$ and the updates of the cluster centers $\{\mu_{\mathcal{A}_i}\}_{i=1}^{K}$. In particular, given $\theta$, we update the gate activation centers by performing K-Means clustering [29] over the gate activations. That is to say, we fix $\theta$ and perform K-means clustering with $K$ clusters over all the gate activations from the dataset $\mathcal{D}$, i.e. we cluster $\mathcal{A}(x_i) \ \forall i$ as shown in Figure 2. Moreover, alternating between optimizing $\mathcal{L}_{\text{DivDR}}$ and updating the gate activation cluster centers over the dataset $\mathcal{D}$, illustrated in Figure 3, results in a diversified set of architectures driven by the data that are more efficient, i.e. learning $K$ local experts that are accurate and efficient.

## 4   Experiments

We show empirically that our proposed DivDR approach can outperform existing methods in better trading off accuracy and efficiency. We demonstrate this on several vision tasks, i.e. semantic segmentation, object detection, and instance segmentation. We start first by introducing the datasets used in all experiments along along with the implementation details. We then present the comparisons between DivDR and several other methods along with several ablations.

### 4.1   Datasets

We mainly prove the effectiveness of the proposed approach for semantic segmentation, object detection, and instance segmentation on two widely-adopted benchmarks, namely Cityscapes [8] and Microsoft COCO [24] dataset.

Table 1: Comparison with baselines on the Cityscapes [8] validation set. * Scores from [21] were reproduced using the official implementation. The evaluation settings are identical to [21]. We calculate the average FLOPs with $1024 \times 2048$ size input.

| Method | Backbone | mIoU$_{val}$(%) | GFLOPs |
|---|---|---|---|
| BiSenet [48] | ResNet-18 | 74.8 | 98.3 |
| DeepLabV3 [5] | ResNet-101-ASPP | 78.5 | 1778.7 |
| Semantic FPN [18] | ResNet-101-FPN | 77.7 | 500.0 |
| DeepLabV3+ [6] | Xception-71-ASPP | 79.6 | 1551.1 |
| PSPNet [49] | ResNet-101-PSP | 79.7 | 2017.6 |
| Auto-DeepLab [25] | Searched-F20-ASPP | 79.7 | 333.3 |
| Auto-DeepLab [25] | Searched-F48-ASPP | 80.3 | 695.0 |
| DR-A [21]* | Layer16 | 72.7±0.6 | 58.7±3.1 |
| DR-B [21]* | Layer16 | 72.6±1.3 | 61.1±3.3 |
| DR-C [21]* | Layer16 | 74.2±0.6 | 68.1±2.5 |
| DR-Raw [21]* | Layer16 | 75.2±0.5 | 99.2±2.5 |
| DivDR-A | Layer16 | 73.5±0.4 | 57.7±3.9 |
| DivDR-Raw | Layer16 | 75.4±1.6 | 95.7±0.9 |

**Cityscapes**. The Cityscapes [8] dataset contains 19 classes in urban scenes, which is widely used for semantic segmentation. It is consist of 5000 fine annotations that can be divided into 2975, 500, and 1525 images for training, validation, and testing, respectively. In the work, we use the Cityscapes dataset to validate the proposed method on semantic segmentation.

**COCO**. Microsoft COCO [24] dataset is a well-known for object detection benchmarking which contains 80 categories in common context. In particular, it includes 118k training images, 5k validation images, and 20k held-out testing images. To prove the performance generalization, we report the results on COCO's validation set for both object detection and instance segmentation tasks.

## 4.2   Implementation Details

In all training settings, we use SGD with a weight decay of $10^{-4}$ and momentum of 0.9 for both datasets. For semantic segmentation on Cityscapes, we use the exponential learning rate schedule with an initial rate of 0.05 and a power of 0.9. For fair comparison, we follow the setting in [21] and use a batch size 8 of random image crops of size $768 \times 768$ and train for $180K$ iterations. We use random flip augmentations where input images are scaled from 0.5 to 2 before cropping. For object detection on COCO we use an initial learning rate of 0.02 and re-scale the shorter edge to 800 pixels and train for 90K iterations. Following prior art, random flip is adopted without random scaling.

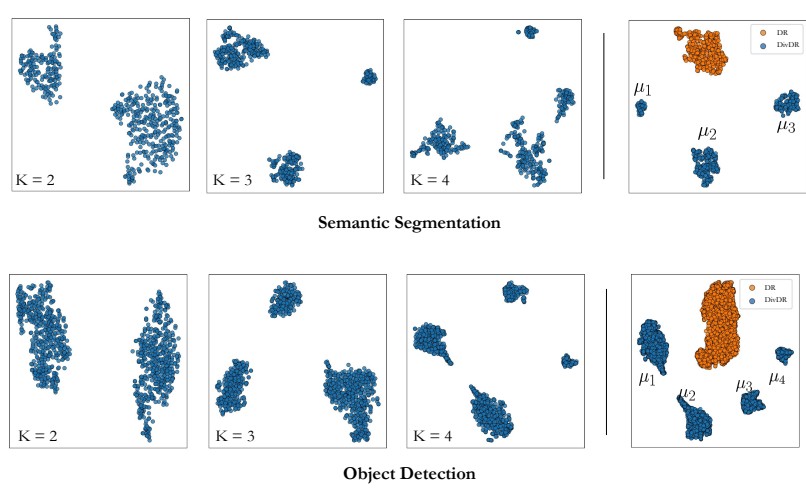

Fig. 4: Visualizing the 183-dimensional $\mathcal{A}$-space of Dynamic Routing backbones trained for semantic segmentation on Cityscapes [8] (*top*) and 198-dimensional $\mathcal{A}$-space for object detection on COCO [24] (*bottom*) using t-SNE [28]. *Left:* varying number of *local experts*, $K = 2, 3, 4$. *Right:* joint t-SNE visualization of architectures of Dynamic Routing [21] (*orange*) and our approach (*blue*). It is clear that our method not only encourages diversity of the learned routes but also reduces variance in a specific cluster. Low *intra*-cluster variance is beneficial because it facilitates feature sharing between similar tasks

## 4.3  Semantic Segmentation

We show the benefits of our proposed DivDR of alternation between training with $\mathcal{L}_{\text{DivDR}}$ and computing the gate activations clusters through K-means on Cityscapes [8] for semantic segmentation. In particular, we compare two versions of our proposed unsupervised Dynamic Routing, namely with and without the computational cost constraint ($\lambda_1 = 0$ denoted as DivDR-Raw and $\lambda_1 = 0.8$ denoted as DivDR-A) against several variants of the original dynamic routing networks both constrained and unconstrained. All experiments are averaged over 3 seeds. As observed in Table 1, while both variants perform similarly in terms of accuracy (DR-Raw: 75.2%, DivDR: 75.4%), DivDR marginally improves the computational cost by 3.5 GFLOPs. On the other hand, when introducing cost efficiency constraint DivDR-A improves both the efficiency (58.7 GFLOPs to 57.7 GFLOPs) and accuracy (72.7% to 73.5%) as compared to DR-A. At last, we observe that comparing to other state-of-the-art, our unconstrained approach, performs similarly to BiSenet [48] with 74.8% accuracy while performing better in computational efficiency (98.3 GFLOPs vs. 95.7 GFLOPs).

Table 2: Quantitative analysis of semantic segmentation on Cityscapes [8]. We report *Inter* and *Intra* cluster variance, that shows how far are the cluster centers are from each other in $L_2$ space and how close are the samples to the cluster centers respectively.

| method | mIoU | FLOPs | Inter | Intra |
|--------|------|-------|-------|-------|
| DR-A | 72.7 | 58.7 | 0.4 | 0.3 |
| DivDR-A | 72.0 | 49.9 | 0.6 | 0.2 |
| DR-Raw | 75.2 | 99.2 | 1.5 | 1.5 |
| DivDR-Raw | 75.7 | 98.3 | 1.2 | 0.5 |

***Visualizing Gate Activations*.** We first start by visualizing the gate activations under different choices of the number of clusters $K$ over the gate activation for DivDR-A. As observed from Figure 4, indeed our proposed $\mathcal{L}_{\text{DivDr}}$ results into clusters on local experts as shown by different gate activations $\mathcal{A}$ for $k \in \{2, 3, 4\}$. Moreover, we also observe that our proposed loss not only results in separated clusters of local experts, i.e. gate activations, but also with a small intra class distances. In particular, as shown in Table 2, our proposed DivDR indeed results in larger inter-cluster distances that are larger than the intra-cluster distnaces. The inter-cluster distances are computed as the average distance over all pair of cluster centers, i.e. $\{\mu_{\mathcal{A}_i}\}_{i=1}^{K}$ while the intra-cluster distances are the average distances over all pairs in every cluster. This indeed confirms that our proposed training approach results in $K$ different architectures for a given dataset. Consequently, we can group the corresponding input images into $K$ classes and visualize them to reveal common semantic features across the groups. For details see Fig 5. We find it interesting that despite we do not provide any direct supervision to the gates about the objects present on the images, the clustering learns to group semantically meaningful groups together.

***Ablating $\alpha$ and $\lambda_2$*.** Moreover, we also ablate the performance of $\alpha$ which is the separation margin in the hinge loss term of our proposed loss. Observe that larger values of $\alpha$ correspond to more enforced regularization on the separation between gate activation clusters. As shown in Figure 6 left, we observe that the mIOU accuracy and the FLOPs of our DivDR-A is only marginally affected by $\alpha$ indicating that a sufficient enough margin can be attained while maintaining accuracy and FLOPs trade-off performance.

## 4.4    Object Detection and Instance Segmentation

To further demonstrate the effectiveness on detection and instance segmentation, we validate the proposed method on the COCO datasets with Faster R-CNN [34] and Mask R-CNN [14] heads. As for the backbone, we extend the original dynamic routing networks with another 5-stage layer to keep consistent with that in FPN [22], bringing 17 layers in total. Similar to that in Sec. 4.3, no external

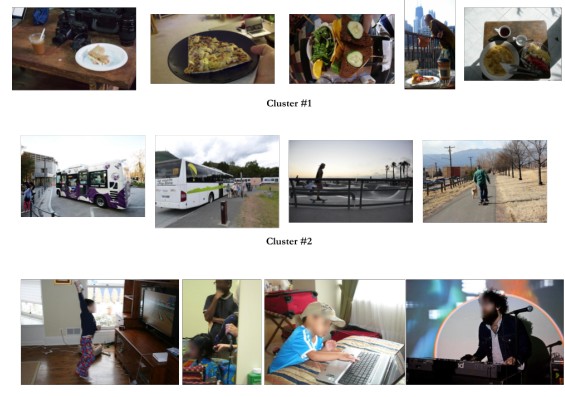

Fig. 5: Visualization of images from the validation set of MS-COCO 2017 [24] challenge. In this training $K = 3$ and we visualize the top-5 images that fall closest to their respective cluster centers $\mu_i$. Note that the dataset does not provide subset-level annotations, however our method uses different pathways to process images containing meals (*top row*), objects with wheels and outdoor scenes (*middle row*) and electronic devices (*bottom row*).

supervision is provided to our proposed DivDR during training. As presented in Tables 4 and 5, we conduct experiments with two different settings, namely without and with computational cost constraints. We illustrate the overall improvement over DR [21] across various hyper-parameters in Fig 8

**Detection.** Given no computational constraints, DivDR attains 38.1% mAP with 32.9 GFLOPs as opposed to 37.7% mAP for DR-R. While the average precision is similar, we observe a noticeable gain computational reduction of 5.3 GFLOPs. Compared with the ResNet-50-FPN for backbone, DivDR achieves similar performance but a small gain of 0.2% but with half of the GFLOPs (32.9 GFLOPs vs. 95.7 GFLOPs). When we introduce the computational regularization, the cost is reduced to 19.8 GFLOPs while the performance is preserved with 35.4% mAP. Compared with that in DR-A, we observe that while Div-DR constraibntconstrainted enjoys a 1.1 lower GLOPS, it enjoys improved precision of 3.3% (35.4% mAP vs. 32.1% mAP) with a lower standard deviation.We believe that this is due to the local experts learnt for separate subsets of the data.

**Instance Segmentation.** As for the task of instance, as observed in Table 5, DivDR unconstrainted performs similarly to DR-R with 35.1% mAP. However, DivDR better trades-off the GLOPs with with a 32.9 GFLOPs in the unconstrained regime as opposed to 38.2 GLOPS. This is similar to the observations made in the detection experiments. Moreover, when computational constraints

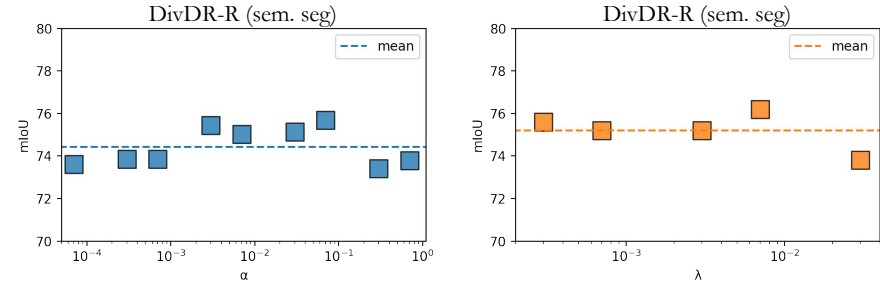

Fig. 6: Ablation on the $\alpha$ (*left*) and $\lambda_2$ (*right*) parameter of the diversity loss term for Semantic Segmentation. The *mean* accuracy in case of the parameter sweep for $\lambda_2$ is higher since in each case the best performing $\alpha$ was used for the training. We can see that the method is stable regardless the choice of the parameters over various tasks.

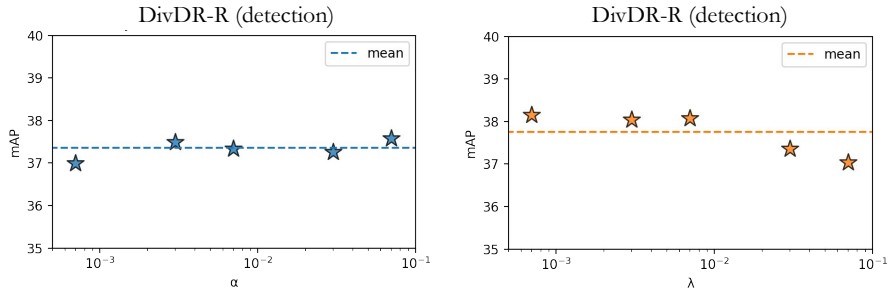

Fig. 7: Ablation on the $\alpha$ (*left*) and $\lambda_2$ (*right*) parameter of the diversity loss term for Object Detection. We can see that the method is stable regardless the choice of the parameters over various tasks.

are introduced, DivDR enjoys a similar GLOPs as DR-A but with an improved 1.6% precision (33.4% mAP vs. 31.8% mAP).

**Ablating $K$.** We compare the performance of our proposed DivDR under different choices of the number of clusters $K$ over the gate activation for both unconstrained and constrained computational constraints, i.e. DivDR-A and DivDR-R respectively. We note that our proposed $\mathcal{L}_{\mathrm{DivDr}}$ effectively clusters the gate activation cluster centers as shown in Figure 4. Moreover, we also observe that our proposed loss not only results in separated clusters of local experts, but also with a small intra-cluster distances as shown in Table 3. In particular, we observe that our proposed DivDR results in larger inter-cluster distances that are larger than the intra-cluster distances (in contrast with DR [21]).

**Ablating $\alpha$ and $\lambda_2$.** As shown in Figure 7, we observe the choice of both both $\alpha$ and $\lambda_2$ only marginally affect the performance of DivDR-A in terms of both

Table 3: Quantitative comparison of Dynamic Routing [21] trained without the objective to diversify the paths and using various $K$ for the clustering term. We omit $K = 1$ from our results as it reverts to forcing the model to use the same architecture, independent of the input image. Instead we report the baseline scores from [21] For comparison we report best Dynamic Routing [21] scores from 3 identical runs with different seeds.

(a) DivDR-A

| K | mAP$_{val}$ | GFLOPs | Inter | Intra |
|---|---|---|---|---|
| * | 34.6 | 23.2 | 0.2 | 0.3 |
| 2 | **35.1** | 21.9 | 1.1 | 0.4 |
| 3 | 35.0 | **19.2** | 0.8 | 0.3 |
| 4 | 34.9 | 20.0 | 0.6 | 0.1 |

(b) DivDR-Raw

| K | mAP$_{val}$ | GFLOPs | Inter | Intra |
|---|---|---|---|---|
| * | 37.8 | 38.2 | 0.5 | 0.7 |
| 2 | 36.5 | **31.0** | 0.6 | 0.5 |
| 3 | 37.4 | 32.6 | 1.2 | 0.5 |
| 4 | **38.1** | 32.8 | 0.7 | 0.2 |

Table 4: Comparison with baselines on the COCO [24] **detection** validation set. * Scores from [21] were reproduced using the official implementation. The evaluation settings are identical to [34] with single scale. We calculate the average FLOPs with $800 \times 800$ size input

| Method | Backbone | mAP$_{val}$ | GFLOPs |
|---|---|---|---|
| Faster R-CNN [34] | ResNet-50-FPN | 37.9 | 88.4 |
| DR-A [21]* | Layer17 | 32.1±5.0 | 20.9±2.1 |
| DR-B [21]* | Layer17 | 36.5±0.2 | 24.4±1.2 |
| DR-C [21]* | Layer17 | 37.1±0.2 | 26.7±0.4 |
| DR-R [21]* | Layer17 | 37.7±0.1 | 38.2±0.0 |
| DivDR-A | Layer17 | 35.4±0.2 | 19.8±1.0 |
| DivDR-R | Layer17 | 38.1±0.0 | 32.9±0.1 |

mAP on the object detection task. However, we find that $\lambda_2 > 0.5$ starts to later affect the mAP for reduced computation.

## 5   Discussion and Future Work

In this paper we demonstrate the superiority of networks trained on a subset of the training set holding similar properties, which we refer to as *local experts*. We address the two main challenges of training and employing local experts in real life scenarios, where subset labels are not available during test nor training time. Followed by that, we propose a method, called Diversified Dynamic Routing that is capable of jointly learning local experts and subset labels without supervision. In a controlled study, where the subset labels are known, we showed that we can recover the original subset labels with 98.2% accuracy while maintaining

Table 5: Comparison with baselines on the COCO [24] **segmentation** validation set. * Scores from [21] were reproduced using the official implementation. The evaluation settings are identical to [34] with single scale. We calculate the average FLOPs with $800 \times 800$ size input

| Method | Backbone | mAP$_{val}$ | GFLOPs |
|---|---|---|---|
| Mask R-CNN [34] | ResNet-50-FPN | 35.2 | 88.4 |
| DR-A [21]* | Layer17 | 31.8±3.1 | 23.7±4.2 |
| DR-B [21]* | Layer17 | 33.9±0.4 | 25.2±2.3 |
| DR-C [21]* | Layer17 | 34.3±0.2 | 28.9±0.7 |
| DR-R [21]* | Layer17 | 35.1±0.2 | 38.2±0.1 |
| DivDR-A | Layer17 | 33.4±0.2 | 24.5±2.3 |
| DivDR-R | Layer17 | 35.1±0.1 | 32.9±0.2 |

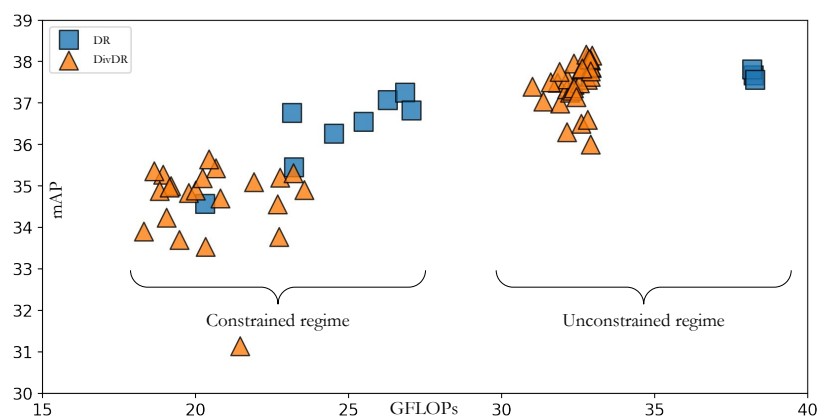

Fig. 8: Evaluations of models trained on COCO [24] across different hyper-parameters

the performance of a hypothetical *Oracle* model in terms of both accuracy and efficiency.

To analyse how well this improvement translates to real life problems we conducted extensive experiments on complex computer vision tasks such as segmenting street objects on images taken from the driver's perspective, as well as detecting common objects in both indoor and outdoor scenes. In each scenario we demonstrate that our method outperforms Dynamic Routing [21].

Even though this approach is powerful in a sense that it could improve on a strong baseline, we are aware that the clustering method still assumes subsets of *equal* and more importantly *sufficient* size. If the dataset is significantly imbalanced w.r.t. local biases the K-means approach might fail. One further limitation is that if the subsets are too small for the *local experts* to learn generalizable rep-

resentations our approach might also fail to generalize. Finally, since the search space of the architectures in this work is defined by Dynamic Routing [21] which is heavily focused on scale-varience. We believe that our work can be further generalized by analyzing and resolving the challenges mentioned above.

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

# 6    Supplementary Material

## 6.1    Sensitivity to number of iterations between K-means update

In our early experiments we have found our method achieving satisfactory results if we kept the number of iterations between the K-means update low: $\leq 100$. With lower frequency updates the diversity between the cluster centers was not sufficiently large, leading to the trivial solution, i.e. the model architecture learning to ignore the input image. In Deep Clustering [2] another technique is mentioned to avoid such trivial solutions, namely randomizing and manually altering the cluster centers in case they happen to be too close to each-other. We did not employ such techniques for our method.

On another note, we have found that while the cluster centers change significantly during the early phases of the training, the difference between two updates is less emphasized towards the end. This lead to a hypothesis that using an annealing policy on the frequency of the updates might be more practical as it could reduce the training time drastically, however such comparison is beyond the scope of this work.

In our experiments we use 50 iterations per K-means update everywhere.

## 6.2    Gathering gate activation values before or after non-linear layer

We have experimented with applying our method on the output of the final linear layer of each gate in our model. We have found that even though much higher variances can be achieved in terms of intra-cluster and inter-cluster diversity metrics, however most of these differences are marginalized by the final non-linear layer of the gates. In the most frequent case the model learned cluster centers that had negative values, which is entirely ignored by the ReLU-part of the non-linear function used by Dynamic Routing [21].