# OpenReview forum: "Diversified Dynamic Routing for Vision Tasks"
_thecvf.com/ECCV/2022/Workshop/VIPriors — VIPriors 2022 OralPosterTBD_

### Official Review · Reviewer_xpff · 2022-08-01
**A well written paper, with some interesting ideas. The results only show marginal improvement across different settings.**

**Rating:** 6
**Confidence:** 4

**Review:**

Summary:
The paper builds on the idea of Dynamic Routing (DR) in the context of mixture of experts. The authors propose an unsupervised DR approach (coined as DivDR) to train several local experts on subsets of a training dataset. The paper is well written and well structured, even though it can benefit from a proof read. The qualitative experimental results are promising and demonstrate the efficacy of the proposed approach in clustering of data subsets and assignment of correct local expats. The impact on the bottomline performance in different settings (object detection, semantic segmentation, and so) is rather marginal. The paper has a coherent story, but lacks solid theoretical deep dive into the mechanics of DivDR.

Major Remarks:
- Any theoretical guarantees, or intuitive analyses, on why the alternating between solving eq(3) and reclustering (3) would work? At least try to discuss this by drawing resemblance with similar approaches in literature.

- Given that there is still enough space, I suggest summarizing the steps involved in Fig 2 and 3 (Subsection 3.2) in the algorithmic form.

- $K$ seems to be an important parameter. How to optimize or tune on this parameter? In Table 3, rather contradicting results are reported. Is it better to increase, decrease or optimize $K$, and how?

- Looking at the result on semantic segmentation, the standard (no-DR based) baselines do not represent the state-of-the-art performance on Cityscapes. That said, the improvement offered by DivDR-X is rather marginal (within 1%). And, the reduction in computational complexity in FLOPS is also on the marginal side. Even though the proposed approach does well in metric learning in the $\mathcal{A}()$ space, the impact on the bottom-line performance seems to be marginal. How would you justify adopting DivDR for semantic segmentation, e.g.? Same can be said about Object detection results, btw.

Minor modifications:
- Another proof read would help to fix typos such as: "evcaluate" (Lines 66-67), and "on subsets on subsets" (Lines 102-3), "of of accuracy" (Line 178)  and so on.

- Please define the acronyms for the first time: NAS (being neural architecture search), etc.

- What is $n$ in Line 167.

-  Please use references here "As shown earlier, learning local experts can benefit performance both in terms of accuracy and computational cost"

---

### Official Review · Reviewer_6cuZ · 2022-08-04
**In general, good work**

**Rating:** 7
**Confidence:** 4

**Review:**

Quality: Good work, intensive experiments with three tasks with two major datasets

Clarity: Clear problem statements, clear and tidy methodology by adding unsupervised k-means and corresponding loss

Significance: the methodology could ease the stated problems to some extend (e.g., if new unseen samples are far away from any local experts, so in this situation the hyper parameter K will be related with the performance. How can you determine this K? And what is the influence? Is there any optimal option on choosing K in general?)

Questions:

1. In Fig. 6 \& 7 right side, the mAP performance dropped with the large \lambda (the rightmost square and star). And the caption for that is 'the method is stable regardless the choice of the parameters over various tasks'. Could you please give me some explanation on those two points? I think the trend of dropping of mAP with large \lambda is clear. Do you still think it is stable?

2. I find some typos or grammar mistakes, please check them. Line 66, 'evcaluate'. Line 112, 'where early on in the training'. Line 236, 'Thereof'. Line 493. 'both both'.

---

### Decision · Program_Chairs · 2022-08-08

**Decision:**

Accept (Oral/Poster TBD)

**Comment:**

Dear authors,


Congratulations! Your work has been accepted to the VIPriors workshop. Decisions on oral/poster presentations will follow later, when the program of the workshop is finalized.

*Please note the first action item is due on Wednesday! Please see instructions below.*

**Camera-ready instructions**

There is some work left to be done to ensure your work is included in the ECCV conference workshop proceedings. The ECCV publication managers use CMT to collect all workshop papers. This means we will migrate your paper from the VIPriors OpenReview page to the centralized ECCV workshop proceedings CMT page. The VIPriors program committee will ensure the details of your work (name, title, email address) are transferred to the CMT page, after which the ECCV proceeding managers will invite you to upload the camera-ready version of your work to the centralized ECCV CMT workshop proceedings page.

Please carefully follow the following instructions:
- **Before August 10th**, ensure that the first author has a CMT account under the same email address as the OpenReview account through which the accepted work was submitted. This account will be used to invite you to upload the camera-ready paper.
- Fill out this form, to inform us that the CMT account is in order: https://docs.google.com/forms/d/e/1FAIpQLSfyAoPv2_srESKaLRHIsHoWe3Fss1Z50ykdH7SzZpenA0m_5g/viewform
- Await instructions from the ECCV publication organizers, sent through CMT, on how to submit your camera-ready paper.
- Submit the camera-ready paper **before August 22nd**. Follow the camera-ready instructions for the main conference: https://eccv2022.ecva.net/submission/call-for-papers/.

**Attending the workshop**

We invite all authors of accepted works to attend the workshop in person on October 24th 2022 at ECCV in Tel Aviv. Please note a conference registration is required to attend the workshop. The workshop will be hybrid, enabling both in-person and remote attendance. We hope all accepted works can be represented in-person by at least one author, but we understand if this is not possible. Remote attendance of the workshop will be possible, though unfortunately there are limits on presenting works remotely: we intend to enable remote oral presentations, but this is not possible for posters.

Please fill out this form *before September 26th* to inform us of your attendance: https://docs.google.com/forms/d/e/1FAIpQLSfqRhdd2pq8t4CC8hL_c8fQo_TWcbzuQH3KGLzKVE36iTW_oQ/viewform.

**Presenting your work at the workshop**

Authors of all accepted papers are invited to present a poster at the workshop. Instructions on poster format will follow at a later date, but we will ask you to print and bring your own poster to the workshop.


For more information, as well as updates on the program of the workshop, keep an eye on our website: https://vipriors.github.io.

We thank you for choosing to submit to our workshop, and we are very much looking forward to hosting you in person in Tel Aviv!


Kind regards,

Robert-Jan Bruintjes
VIPriors program committee